# Nonstationary Latent Bandits

## Abstract

Addressing non-stationarity and latent variables in bandit algorithms presents significant challenges. This paper tackles both challenges simultaneously in Multi-Agent Multi-Armed Bandits by integrating causal inference principles with panel data methodologies. We propose Dynamic Causal Multi-Armed Bandits (DCMAB) and Dynamic Causal Contextual Bandits (DCCB), focusing on treatment effect estimation rather than direct reward modeling. Our algorithms, employing matrix completion on agent-time reward matrices, effectively leverage shared information among agents. The proposed methods assume either time-invariant or linear treatment effect models, while accommodating flexible data-generating processes for potential outcomes. We establish sub-linear dynamic regret for the proposed algorithms and extend their applicability to scenarios with time-varying treatment effects. Through extensive simulations and a real-world application in the stock market, we validate the superiority of our proposed methods in non-stationary bandits with latent variables.

## 1 Introduction

Bandit algorithms form a fundamental part of reinforcement learning and have found wide application in fields like precision medicine (Lu et al., 2021b; Liao et al., 2020), education (Rafferty et al., 2019; Kizilcec et al., 2020), online recommendation systems (Li et al., 2010; Abel et al., 2011), and dynamic pricing strategies (Xu et al., 2016; Turvey, 2017). In many cases, multiple agents may interact with the environment concurrently, known as Multi-Agent Multi-Armed bandits (MAMAB). At each time point, each agent observes information about the environment, chooses an action based on the information, and then receives feedback, known as reward. Since the agents may have similar reward patterns, leveraging shared information can enhance the learning process and minimize overall regret. Below are two examples illustrating the application of multi-agent bandits:

**Example 1: Stock Market** Athey et al. (2021) Consider a situation where each stock represents an agent, and the goal is to maximize the overall return on investment (ROI). At each time point, the agents evaluate the updated returns for all tradeable stocks and then adjust strategies accordingly. The aim is to find an optimal policy for market interactions to maximize ROI.

**Example 2: Online movie recommendation (Jain & Pal, 2022; Wan et al., 2021)** In online movie streaming platforms, recommendations play a crucial role. When users seek movie suggestions, the platform acts as a decision-maker based on the observed user features. After implementing a specific recommendation policy, the platform gathers user feedback (e.g., clicks, watching duration, ratings), which serves to inform recommendation decisions in future rounds and thereby fosters a dynamic learning environment. In this task, the aim is to enhance user satisfaction by recommending personalized movies based on individual tastes and treating each user as a unique agent.

Despite extensive research in MAMAB (Gentile et al., 2014; Shahrampour et al., 2017; Bargiacchi et al., 2018; Verstraeten et al., 2020), most existing work focuses on information sharing but overlooks two significant challenges: the dynamic nature of rewards and latent variables.

**Challenge 1 Dynamic Nature of Rewards** The first challenge arises from the dynamic nature of rewards. In numerous real-world applications, the reward distributions associated with each arm are rarely static; they evolve, often influenced by various external factors and internal dynamics. For instance, in stock markets

(Example 1), shifts in the economic or political landscape can significantly change the patterns of daily returns. Similarly, in online movie recommendations (Example 2), users' evolving preferences can impact feedback. These changes require continuous adaptation in bandit algorithms to effectively respond to the ever-changing reward distributions.

**Challenge 2 Latent Variables** The second challenge in practical applications involves unobservable latent variables affecting rewards. For example, in stock market analysis (Example 1), latent variables could include unmeasured investor sentiments or undisclosed financial events impacting stock prices. In movie recommendations (Example 2), latent variables might include a user's mood or unmeasured cultural influences affecting their preferences.

In real-life scenarios, the above mentioned challenges often coexist, yet literature addressing both challenges remains limited. Overlooking non-stationary or latent variables in bandit algorithms can lead to misspecified reward modeling and a regret lower bound of $\mathcal{O}(T)$ (Wang et al., 2024; Foster et al., 2020). This emphasizes the critical necessity for innovative approaches that simultaneously address both. Previous studies, like those by Hong et al. (2020) and Nelson et al. (2022), attempt to tackle these challenges but depend on strong assumptions like Markov assumptions or discretized latent states, restricting their general application. Our paper seeks to overcome these limitations by proposing a more flexible approach to the multi-agent, non-stationary latent bandit problem, avoiding restrictive assumptions on non-stationarity and the transitions of latent variables.

**Contributions** Our contributions are fourfold:

First, we are one of the early works that bridge the fields of causal inference (Splawa-Neyman et al., 1990; Rubin, 1974) and bandits. We express implicit assumptions in bandit algorithms using causal inference terminology, thereby establish a framework for identifying treatment effects. We observe that the historical agent reward data, forming an agent-time reward matrix, closely aligns with the structure of panel data which has been well studied in causal inference to estimate treatment effects with the presence of non-stationarity and latent variables in static settings (Abadie et al., 2010; Abadie, 2005; Athey et al., 2021). Yet, extending these methods to dynamic online environments remains less explored. Moreover, we recognize that it is not necessary to estimate the full reward functions for each arm; instead, it suffices to estimate the differences in rewards between arms. This concept aligns closely with the idea of treatment effect estimation in causal inference. These insights motivate us to adapt panel data methods for addressing non-stationarity and latent variables in bandit algorithms.

Second, we introduce two innovative bandit algorithms, namely, Dynamic Causal Multi-Armed Bandits (DCMAB) and Dynamic Causal Contextual Bandits (DCCB), to address latent variables and non-stationarity. These algorithms adopt a flexible reward model estimation strategy by shifting focus from direct reward modeling to treatment effect estimation, assuming either time-invariant treatment effects or linear treatment effects. By employing matrix completion on agent-time reward matrices, our approach efficiently utilizes shared information across agents over time, enabling a more comprehensive approach to decision-making in dynamic scenarios. To the best of our knowledge, our work is the first to establish sub-linear dynamic regret in the literature on policy learning that uses treatment effect estimation in non-stationary settings with latent variables.

Third, to allow greater reward modeling flexibility, we extend DCCB to accommodate time-varying treatment effects. We propose Discounted Dynamic Causal Contextual Bandits (D-DCCB), incorporating a weighted strategy. By adopting a variation budget, a standard assumption in non-stationary bandit research, we ensure that D-DCCB successfully achieves sub-linear dynamic regret.

Finally, we demonstrate the superior performance of our algorithms in non-stationary scenarios with latent variables through extensive simulation studies and a real-world application in the stock market.

## 2 Related Work

**Non-stationary latent bandits** Non-stationary Multi-Armed bandit (MAB) (see e.g., Garivier & Moulines, 2011; Trovo et al., 2020; Cai et al., 2021; Jia et al., 2023; Liu et al., 2023; Chen et al., 2024)

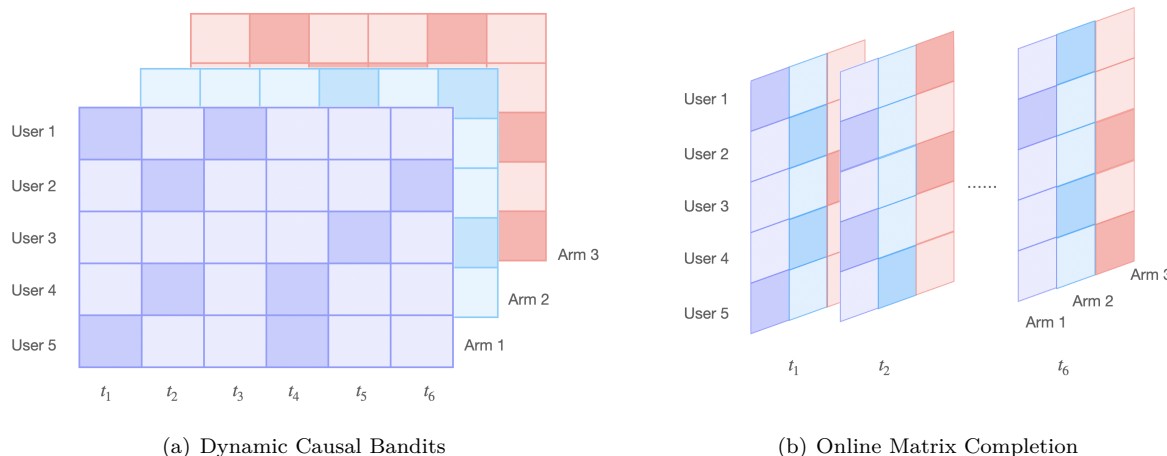

(a) Dynamic Causal Bandits

(b) Online Matrix Completion

Figure 1: Dynamic Causal Bandits vs. Online Matrix Completion: Dark colors represent observed rewards, and light colors indicate unobserved rewards.

and latent MAB (Maillard & Mannor, 2014; Gopalan et al., 2016; Hong et al., 2022) have both received increasing interest in the literature. However, research that simultaneously addresses non-stationary and latent variables remains sparse. Previous studies model the reward evolution process using Markov Chains, focusing on discrete rewards and discrete latent variables (Hong et al., 2020; Nelson et al., 2022; Galozy & Nowaczyk, 2023; Dzhoha & Rozora, 2023; Russo et al., 2023). Yet, the Markovian assumption may not always reflect the complexities of real-world data, and latent variables might exhibit continuous dynamics that escape simple categorization. In such scenarios, existing algorithms may yield suboptimal decisions. In contrast, our paper allows a more flexible setting where latent variables are not necessarily discrete, and Markovian properties may be violated.

**Multi-Agent Multi-Armed bandits with heterogeneous feedback** Research in MAMAB involving heterogeneous feedback is rapidly growing (see e.g., Gentile et al., 2014; Li et al., 2016; Shahrampour et al., 2017). One popular technique is matrix completion, which posits a low-rank structure in the agent-arm reward matrix (see e.g., Févotte et al., 2009; Guan et al., 2012; Wang et al., 2016; Katariya et al., 2017; Kveton et al., 2017; Dadkhahi & Negahban, 2018; Trinh et al., 2020; Lu et al., 2021a; Jain & Pal, 2022). Our proposed methods differ in several ways: First, we remove the cluster structure requirement and allow for diverse reward distributions across different units, accommodating greater agent heterogeneity; Second, Instead of concentrating on agent-arm matrices, we apply matrix completion to agent-time reward matrices, as shown in Figure 1, employing causal inference to address correlations between arms; Last but not least, our model accounts for non-stationary reward distributions that exhibit time-dependent variations, addressing a key complexity in dynamic environments.

**Causal inference in non-stationary bandits** Causal inference (Splawa-Neyman et al., 1990; Rubin, 1974) has recently found its application in bandits algorithms. Sawant et al. (2018) was among the first to integrate causal effect estimators in online bandits. They developed a method for automated marketing that deals with latent variables but without theoretical backing. Zhang et al. (2022) later proposed to address the issue of confounding using instrumental variables. However, research simultaneously addressing non-stationarity and latent variables in bandits remains limited. Carranza et al. (2023) applied the R-Learner (Nie & Wager, 2021) to estimate the Heterogeneous Treatment Effect in panel data while updating policy at each time point. Their approach adopts a restart technique to deal with non-stationarity. Yet, their proposed algorithm can only achieve suboptimal linear regret. Farias et al. (2022) introduced an adaptive experiment design using bandits. Their work focuses on treatment effects estimation rather than policy learning and requires a constant control group, leading to suboptimal regret. To the best of our knowledge, our proposed method is the first to achieve sublinear regret in policy learning that addresses non-stationarity and latent variables simultaneously.

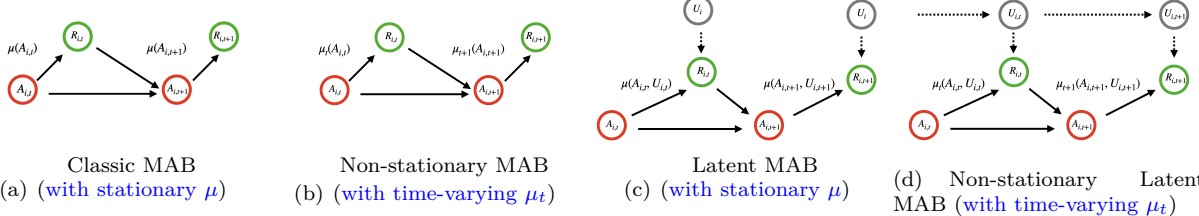

Figure 2: Causal Diagrams. Denote $A_{i,t}$ as the action taken (represented by red circles), $R_{i,t}$ as the reward (in green circles), $U_{i,t}$ as the latent variable (unobserved in grey circles), and $\mu(\cdot)$ as the reward mean function. Subscript $t$ for $\mu(\cdot)$ indicates that the reward mean may change over time. Solid arrows indicate observed causal relationships, while dashed arrows indicate unobserved ones.

Additional literature review on non-stationary bandits, bandits with latent variables and bandits with Causal Structure Models (see e.g., Pearl, 2010) can be found in Appendix A.

## 3 Framework

Consider a Multi-Agent Multi-Armed bandit with a total of $N$ agents ($i = 1, 2, \cdots, N$). For each agent $i$, at each time point $t \in \{1, 2, \cdots, T\}$, a decision-maker takes an action $a_{i,t} \in \mathcal{A}$ based on a set of learned rules, known as policy. Upon taking the action, the decision-maker observes a reward $r_{i,t} \in \mathcal{R}$. Subsequently, the learned policy $\widehat{\pi}_{i,t}$ is updated while interacting with the environment. Let $\mathcal{H}_{i,t-1} = \{\boldsymbol{x}_{i,j}, a_{i,j}, r_{i,j}\}_{1 \leq j \leq t-1}$ denote the history of agent $i$ up to time $t$. Throughout this paper, we use uppercase and lowercase letters to represent random variables and their realizations, respectively, with boldface notation for vectors. We adopt the potential outcome framework in Causal Inference (Splawa-Neyman et al., 1990; Rubin, 1974), where $R_{i,t}(a)$ denotes the potential outcome of agent $i$ at time point $t$ under action $a \in \mathcal{A}$. And we denote $\mathbf{I}$ as an indicator function such that $\mathbf{I}\{a_{i,t} = a\} = 1$ if $a_{i,t} = a$ and 0 otherwise. Throughout the paper, we use capital letters (e.g., $\{\boldsymbol{X}_{i,j}, A_{i,j}, R_{i,j}\}$) to denote random variables and lowercase letters (e.g., $\{\boldsymbol{x}_{i,j}, a_{i,j}, r_{i,j}\}$) to denote their realizations. We follow a delimiter nesting convention: parentheses $(\cdot)$ for the innermost expressions, braces $\{\cdot\}$ for the next level, and brackets $[\cdot]$ for the outermost. If additional levels are needed, we cycle back to parentheses.

For reward modeling, we define the mean reward of agent $i$ at time point $t$ given $a_{i,t} = a$ as $\mu_{i,t}(a) = \mathbb{E}(R_{i,t} \mid a)$. As such, the reward can be expressed as $R_{i,t}(a) = \mu_{i,t}(a) + \epsilon_{i,t}$, where the noise term $\epsilon_{i,t}$ has mean zero. Classical multi-armed bandits assume the reward mean $\mu_{i,t}(a)$ to be constant across time for each action, i.e., $\mu_{i,t}(a) = \mu_i(a)$, as shown in Figure 2(a). However, as discussed in Section 1, the reward distribution often evolves over time, as illustrated in Figure 2(b). Additionally, there may exist a latent variable $\boldsymbol{U}_t$ influencing both the reward and subsequent actions, as shown in Figure 2(c). This paper considers a more general case where a latent variable $\boldsymbol{U}_t$ might exist and the reward distribution changes over time, as depicted in Figure 2(d).

The goal of classical bandit algorithms is to learn a policy that maximizes cumulative rewards. At each time point $t$, the decision-maker estimates the reward mean function $\widehat{\mu}_{i,t}$ for each agent and action. Based on this estimation, the optimal policy is estimated as $\widehat{\pi}_{i,t} = \arg\max_{a \in \mathcal{A}} \widehat{\mu}_{i,t}(a)$. Subsequently, an action $a_{i,t}$ is selected based on the estimated optimal policy $\widehat{\pi}_{i,t}$. We define the optimal policy as $\pi_{i,t}^* \equiv \arg\max_{a \in \mathcal{A}} \mu_{i,t}(a)$. And we measure the dynamic regret to evaluate the learning performance, which is the difference between the reward under the optimal policy and the chosen action, defined as $\Delta_{i,t} = r_{i,t}(\pi_{i,t}^*) - r_{i,t}(a_{i,t})$. Our goal is to learn the optimal policy while minimizing the cumulative dynamic regret, summed over all agents and all time points, defined as $\mathcal{R}_{N,T} = \sum_{i=1}^{N} \sum_{t=1}^{T} \Delta_{i,t}$.

## 4 Dynamic Causal Bandits

In this section, we propose Dynamic Causal Bandits by incorporating causal effects. Without loss of generality, we consider two–armed bandits with $\mathcal{A} = \{0, 1\}$ for simplicity and clarity.

### 4.1 Causal Effect Identification in Bandits

We first bridge the fields of bandit algorithms and Causal Inference by establishing the causal effect identification in bandits. We assume that the observed outcome of a particular agent depends solely on the action it takes, unaffected by the actions chosen by other agents, stated as follows:

**A 1.** (Consistency) For any $i \geq 1$, $t \geq 1$, assume $R_{i,t} = \sum_{a \in \mathcal{A}} \mathbf{I}\{A_{i,t} = a\} R_{i,t}(a)$.

A1 uses the terminologies and concepts from Causal Inference, and is often implicitly assumed in the literature of Multi-Agent Bandits (see e.g., Guan et al., 2012; Gentile et al., 2014; Kawale et al., 2015; Wang et al., 2016; Sankararaman et al., 2019; Trinh et al., 2020; Lu et al., 2021a; Wang et al., 2021) except in the case of Multi-Agent Bandits with collisions (see e.g., Liu & Zhao, 2010; Kalathil et al., 2014).

And we require that the decision-maker does not influence the evolution of the rewards. This assumption can be expressed as conditional independence using the language of Causal Inference:

**A 2.** (Conditional Independence) For any $i \geq 1$, $t \geq 1$, and $a \in \mathcal{A}$, we have $R_{i,t}(a) \perp A_{i,t} \mid \mathcal{H}_{i,t-1}$.

A2 is a bandit version of Sequential Randomization Assumption considered in Dynamic Treatment Regimes (Robins, 1987) required for potential outcome identification. This assumption is inherently met in classical stationary bandits since the rewards follow an i.i.d. distribution that remains consistent over time. However, in non-stationary bandits, this assumption imposes additional requirements. As explored by Tekin & Liu (2012) and others, non-stationary bandits can be divided into two categories: rested and restless. Rested bandits feature static reward distributions for each arm until played. In contrast, restless bandits, which are common and realistic models of many real-world scenarios, have reward distributions that continuously evolve, regardless of player actions. Our focus in this paper is on the non-stationary nature of rewards due to dynamic environments, typical of the restless case, thereby satisfying A2.

It's important to note that A2 does not exclude the existence of latent variables. This assumption is widely adopted in many bandit works that involve latent variables (see e.g., Greenewald et al., 2017; Krishnamurthy et al., 2018; Ou et al., 2019; Kim & Paik, 2019; Peng et al., 2019; Choi et al., 2022) and bandits using causal inference (Sawant et al., 2018; Carranza et al., 2023; Farias et al., 2022). There may be latent variables affecting both potential outcomes and historical data, which are then used to learn the policy and select actions in subsequent time points. However, with the given historical context $\mathcal{H}_{i,t-1}$, the action $A_{i,t}$ is determined by the learning algorithm and remains independent of the reward $R_{i,t}(a)$.

With A1 and A2, the potential outcome at each time point $t$ is identifiable given $\mathcal{H}_{i,t-1}$, as

$$\mathbb{E}\{R_{i,t}(a) \mid \mathcal{H}_{i,t-1}\} \overset{A2}{=} \mathbb{E}\{R_{i,t}(a) \mid A_{i,t} = a, \mathcal{H}_{i,t-1}\}$$
$$\overset{A1}{=} \mathbb{E}(R_{i,t} \mid A_{i,t} = a, \mathcal{H}_{i,t-1}).$$

Thus, we are able to estimate the individual treatment effect using observed history data in bandits:

$$\begin{aligned}
\tau_{i,t} := & \mathbb{E}\{R_{i,t}(1) - R_{i,t}(0) \mid \mathcal{H}_{i,t-1}\} \\
= & \mathbb{E}(R_{i,t} \mid A_{i,t} = 1, \mathcal{H}_{i,t-1}) \\
& - \mathbb{E}(R_{i,t} \mid A_{i,t} = 0, \mathcal{H}_{i,t-1}).
\end{aligned} \tag{1}$$

### 4.2 Dynamic Causal Bandits Algorithms

In this section, we propose Dynamic Causal Bandit algorithms that optimize policies based on estimated causal effects—rather than potentially biased direct reward observations—in order to maximize cumulative

---

**Algorithm 1** Dynamic Causal Multi-Armed Bandits (DCMAB)

---

**Input:** Burning period $t_0$; Number of agents $N$; Epsilon-Greedy parameter $\epsilon_t$.
**for** Time $t = 1, \cdots, t_0$ **do**
    $a_{i,t} \sim \text{Bernoulli}(0.5), 1 \leq i \leq N$;
**end for**
**for** Time $t = t_0 + 1, \cdots, T$ **do**
    **for** Arm $a \in \{0, 1\}$ **do**
        **Step 1:** Use equation 3 to estimate $\widehat{\mathbf{M}}_{t-1}(a) = \{\widehat{\mu}_{i,s}(a)\}_{1 \leq i \leq N, 1 \leq s \leq t-1}$.
    **end for**
    **for** Agent $i = 1, 2, \cdots, N$ **do**
        **Step 2:** Estimate $\widehat{\tau}_{i,s} = \widehat{\mu}_{i,s}(1) - \widehat{\mu}_{i,s}(0)$ for $s = 1, 2, \cdots, t-1$;
        **Step 3:** Estimate $\widetilde{\tau}_{i,t} = \frac{1}{t-1} \sum_{s=1}^{t-1} \widehat{\tau}_{i,s}$;
        **Step 4:** Estimate the optimal arm $\widehat{\pi}_{i,t} = \mathbf{I}\{\widetilde{\tau}_{i,t} > 0\}$;
        **Step 5:** Choose arm $a_{i,t} = \widehat{\pi}_{i,t}$ with probability $(1 - \epsilon_t)$, and choose arm $a_{i,t} \sim \text{Bernoulli}(0.5)$ with
probability $\epsilon_t$;
        **Step 6:** Receive reward $r_{i,t}$;
    **end for**
**end for**

---

rewards. Our methods innovatively apply matrix completion to agent-time reward matrices, represented as $\mathbf{R}_t(a) = \{R_{i,s}(a)\}_{1 \leq i \leq N, 1 \leq s \leq t}$ for each arm as illustrated in Figure 1. We model the potential outcome of these reward matrices for each arm at each time point $t$ as:

$$\mathbf{R}_t(a) = \mathbf{M}_t(a) + \mathbf{E}_t(a), \tag{2}$$

where $\mathbf{M}_t(a) = \{\mu_{i,s}(a)\}_{1 \leq i \leq N, 1 \leq s \leq t}$ and $\mathbf{E}_t(a) = \{\varepsilon_{i,s}(a)\}_{1 \leq i \leq N, 1 \leq s \leq t}$, with $\varepsilon_{i,s}(a)$ being $\sigma_a$-sub-Gaussian, independent of each other and independent of $\mathbf{M}_t(a)$. Denote $S_a(N, t)$ as the rank of the reward matrix $\mathbf{M}_t(a)$. Denote $\mathbf{W}_t(a) = \{\mathbf{I}\{a_{i,s} = a\}\}_{1 \leq i \leq N, 1 \leq s \leq t}$ as the indicator matrix. The matrix completion process involves minimizing the sum of squared differences:

$$\widehat{\mathbf{M}}_t(a) = \arg\min_{\mathbf{M}} \frac{1}{Nt} \|\boldsymbol{W}_t(a) \circ (\mathbf{M} - \mathbf{R}_t(a))\|_F^2 + \lambda_t \|\mathbf{M}\|_* \tag{3}$$

where $\circ$ denote the Hadamard product, $\lambda_t$ is a pre-selected penalty parameter, $\|\mathbf{M}\|_F$ denotes the Frobenius norm of $\mathbf{M}$, and $\|\mathbf{M}\|_*$ denotes the nuclear norm of $\mathbf{M}$.

**Dynamic Causal Multi-Armed Bandits (DCMAB)** In the case that the individual treatment effect remains consistent for each unit over time, we propose Dynamic Causal Multi-Armed Bandits (DCMAB), as outlined in Algorithm 1. At each iteration, DCMAB first uses matrix completion as detailed in equation 3, to reconstruct the complete agent-time reward matrices in Step 1. This step is crucial for learning the counterfactual outcomes. Subsequently, in Step 2, the algorithm estimates the individual treatment effect for each unit at every time point, $\widehat{\tau}_{i,s} = \widehat{\mu}_{i,s}(1) - \widehat{\mu}_{i,s}(0)$. Then in Step 3, we use sample average over time $\widetilde{\tau}_{i,t} = \frac{1}{t-1} \sum_{s=1}^{t-1} \widehat{\tau}_{i,s}$ to estimate the treatment effect for each unit, which is subsequently used to estimate the optimal arm $\widehat{\pi}_{i,t}$ in Step 4. The algorithm then optimizes decision-making by implementing a classic epsilon-greedy strategy in Step 5. Our algorithm effectively captures latent structure in counterfactual outcomes through matrix completion, which might not be immediately evident from direct reward observations. By incorporating concepts of causal effects, DCMAB deduces the underlying dynamics of the environment and focuses on the time-invariant treatment effects, thus effectively addressing the challenges of non-stationarity and latent variables simultaneously.

**Dynamic Causal Contextual Bandits (DCCB)** When contextual information $\boldsymbol{x}_{i,t} \in \mathcal{R}^d$ is available for each unit at each time point, we can extend our approach to Dynamic Causal Contextual Bandits (DCCB). We model the treatment effect $\tau_{i,t}$ as a linear function of the context variables, denoted as $\tau_{i,t} = \boldsymbol{x}_{i,t}' \boldsymbol{\beta}_i$, where $\boldsymbol{\beta}_i$ is a vector of coefficients assumed to remain constant over time for each individual unit $i$. At each time point $t$, we first estimate the individual treatment effect of each unit $\widehat{\tau}_{i,t}$ using Step 1-2 in Algorithm

---

**Algorithm 2** Dynamic Causal Contextual Bandits (DCCB)

---

**Input:** Burning period $t_0$; Number of agents $N$; Epsilon-Greedy parameter $\epsilon_t$.
Burning period $t = 1, \cdots, t_0$: same as Algorithm 1
**for** Time $t = t_0 + 1, \cdots, T$ **do**
    **for** Arm $a \in \{0, 1\}$ **do**
        **Step 1:** Use equation 3 to estimate $\widehat{\mathbf{M}}_{t-1}(a)$.
    **end for**
    **for** Agent $i = 1, 2, \cdots, N$ **do**
        **Step 2:** Estimate $\widehat{\tau}_{i,s} = \widehat{\mu}_{i,s}(1) - \widehat{\mu}_{i,s}(0)$ for $s = 1, 2, \cdots, t-1$;
        **Step 3:** Fit OLS estimator $\widehat{\boldsymbol{\beta}}_{i,t-1} = \left\{ \sum_{s=1}^{t-1} \boldsymbol{x}_{i,s} \boldsymbol{x}'_{i,s} \right\}^{-1} \sum_{s=1}^{t-1} \boldsymbol{x}_{i,s} \widehat{\tau}_{i,s}$, and estimate the treatment
effect as $\widetilde{\tau}_{i,t} = \boldsymbol{x}'_{i,t} \widehat{\boldsymbol{\beta}}_{i,t-1}$;
        **Step 4:** Estimate the optimal arm $\widehat{\pi}_{i,t} = \mathbf{I}\{\widetilde{\tau}_{i,t} > 0\}$;
        **Step 5:** Choose arm $a_{i,t} = \widehat{\pi}_{i,t}$ with probability $(1 - \epsilon_t)$, and $a_{i,t} \sim \text{Bernoulli}(0.5)$ with probability
$\epsilon_t$;
        **Step 6:** Receive reward $r_{i,t}$;
    **end for**
**end for**

---

1. Then instead of using the sample average as Step 3 in Algorithm 1, in DCCB, we regress $\widehat{\tau}_{i,t}$ on the contextual information $\boldsymbol{x}_{i,t}$ to estimate the covariates $\widehat{\boldsymbol{\beta}}_i$ using Ordinary Least Squares (OLS). The OLS estimation is formulated as: $\widehat{\boldsymbol{\beta}}_{i,t-1} = \arg\min_{\boldsymbol{\beta} \in \mathcal{R}^d} \sum_{s=1}^{t} \left( \widehat{\tau}_{i,s} - \boldsymbol{x}'_{i,s} \boldsymbol{\beta} \right)^2 = \left\{ \sum_{s=1}^{t-1} \boldsymbol{x}_{i,s} \boldsymbol{x}'_{i,s} \right\}^{-1} \sum_{s=1}^{t-1} \boldsymbol{x}_{i,s} \widehat{\tau}_{i,s}$.

Then utilizing the estimated treatment effects $\widetilde{\tau}_{i,t} = \boldsymbol{x}'_{i,t} \widehat{\boldsymbol{\beta}}_{i,t-1}$, the algorithm determines the optimal action for each unit following the epsilon-greedy strategy. The DCCB algorithm is outlined in Algorithm 2. By leveraging context-specific information, this algorithm refines its treatment effect estimations, leading to more informed decisions in dynamic environments.

*Remark* 1. The epsilon-greedy strategy is chosen to ensure sufficient exploration at every time point from a theoretical perspective. For alternative strategies like Upper Confidence Bound and Thompson Sampling, we could employ a clipping technique as discussed in Zhang et al. (2020) and Shen et al. (2024) to ensure sufficient exploration.

*Remark* 2. Algorithms for k arms can be found in Appendix B.

## 5 Regret Analysis

In this section, we establish theoretical guarantees for the proposed DCMAB and DCCB algorithms. Specifically, we establish upper bounds on the accumulated dynamic regret of both algorithms when $T < N$. We remark that in real-life scenarios, especially in the early stages of data collection or in environments with a limited operational duration, it is common to encounter situations where $T < N$ (Jain & Pal, 2022; Pal et al., 2023; Farias et al., 2022).

**Theorem 1** (Regret Bound for DCMAB). *Under A1, A2, A4 - 6 in the Appendix, and assume $\tau_{i,t} = \tau$ is a constant overtime, $S(N) \triangleq \max_{a \in \mathcal{A}} \left\{ \sigma_a^2 S_a(N, t) \right\}$ remains constant as $T$ increases, provided $T < N$ with appropriately selected $\lambda_t$ and $\epsilon_t = c \frac{1}{N^\omega t^\alpha \sqrt{\log(N+t)}}$ for constant $c > 0$, and $0 < \omega, \alpha < 1/2$ , the dynamic regret of Algorithm 1 is bounded by $\mathcal{O}\left( \frac{N^{1-\omega} T^{1-\alpha}}{\sqrt{\log(N+T)}} + S(N) \log(N) N^{2\omega} T^{1/2+\alpha} \right)$.*

The detailed proof is available in Appendix D.2. Assumptions A4 and A5 are standard boundedness conditions to prevent extreme reward values. Assumption A6 is a commonly used margin condition in the causal inference literature. To establish the regret bound, we decompose the expected regret $\mathbb{E}\{\mathcal{R}_{N,T}\}$ into two principal components: $\mathbb{E}\{\mathcal{R}_{N,T}\} = \sum_{t=1}^{T} N\epsilon_t + \sum_{t=1}^{T} \sum_{i=1}^{N} \Pr\left( \widehat{\pi}_{i,t} \neq \pi_{i,t}^* \right)$. The first part $\sum_{t=1}^{T} N\epsilon_t$ is due to exploration using epsilon-greedy strategy in the learning process. The second part, $\sum_{t=1}^{T} \sum_{i=1}^{N} \Pr\left( \widehat{\pi}_{i,t} \neq \pi_{i,t}^* \right)$,

results from errors in estimating the treatment effects. This component accounts for two scenarios: one where all treatment effects for each unit are close to zero with probability bounded by $\mathcal{O}\left(S(N)\log(N)\sqrt{T}\right)$, and the other where not all treatment effects are negligible, in which case the matrix completion estimation error is bounded by $\mathcal{O}\left((N+T)^{-1}\right)$. Overall, the dynamic regret resulting from estimation error in the treatment effects is small. This theorem reveals DCMAB's efficiency in minimizing dynamic regret through both controlled exploration and precise treatment effect estimations, demonstrating its capability in managing dynamic multi-agent environments with latent variables.

When context $\boldsymbol{x}_{i,t}$ is available, we further have:

**Theorem 2** (Regret Bound for DCCB). *Assume the conditions in Theorem 1 hold with $\tau_{i,t} = \boldsymbol{x}'_{i,t}\boldsymbol{\beta}_i$ and A7 - 8 in Appendix, we have the dynamic regret of Algorithm 2 is bounded by* $\mathcal{O}\left(\frac{N^{1-\omega}T^{1-\alpha}}{\sqrt{\log(N+T)}} + S(N)\log(N)N^{2\omega}T^{1/2+\alpha}\right).$

The proof of Theorem 2 is similar to that of Theorem 1, with the exception that it accounts for the estimation error from OLS estimators. To maintain clarity and focus on the scaling behavior with respect to the number of agents $N$ and time horizon $T$, we treat the context dimension $d$ as fixed in the main text and defer detailed dependence on $d$ to the detailed proof in appendix D.3.

We remark that we are the first to derive a sublinear dynamic regret for policy learning in non-stationary latent bandits with causal inference. While Sawant et al. (2018) pioneered the integration of single-stage treatment effect estimators in an online bandit setting, they did not provide theoretical analysis. On the other hand, Carranza et al. (2023) could only achieve linear regret in $T$. In comparison with the literature on MAMAB with heterogeneous feedback, the only algorithm that allows heterogeneous feedback within clusters attains a regret bound of $\mathcal{O}\left(N\sqrt{T\log(T)}\right)$(Wang et al., 2021), which is suboptimal in $N$. Although our regret bound grows faster with $T$, our approach accounts for the non-stationarity of rewards, a factor not considered by Wang et al. (2021), who assume stationary rewards over time. Overall, our approach attains a dynamic regret bound that is sublinear in both $N$ and $T$, resulting in a more favorable regret performance.

## 6 Extensions

In this section, we explore extensions of the DCCB algorithm to handle scenarios involving non-stationary treatment effects for greater reward modeling flexibility. We model the treatment effect $\tau_{i,t}$ as a linear function of the context variables, represented as $\tau_{i,t} = \boldsymbol{x}'_{i,t}\boldsymbol{\beta}_{i,t}$. Different from the model used in DCCB, here $\boldsymbol{\beta}_{i,t}$ is subject to change over time independently for each individual unit $i$. Instead of OLS, we propose the implementation of weighted least squares Russac et al. (2019); Wang et al. (2023) as a method to progressively discount past rewards, $\widehat{\boldsymbol{\beta}}_{i,t-1} = \arg\min_{\boldsymbol{\beta}\in\mathcal{R}^d}\sum_{s=1}^{t}\rho^{t-1-s}\left(\widehat{\tau}_{i,s} - \boldsymbol{x}'_{i,s}\boldsymbol{\beta}\right)^2$, where $\rho\in(0,1)$ is a use-defined constant. We summarize our proposed Discounted Dynamic Causal Contextual Bandits(D-DCCB) in Algorithm 3. This approach is designed to assign more weight to recent data, thereby enhancing the algorithm's adaptability to evolving environments.

With an additional assumption on variation budget widely used in non-stationary bandit literature (see e.g., Besbes et al., 2015; 2014; Russac et al., 2019; Wang et al., 2023), we are able to derive the dynamic regret bound for Algorithm 3 as follows.

**A 3** (variation budget). For any $i$, denote $B_t = \sum_{p=1}^{t}\|\boldsymbol{\beta}_p - \boldsymbol{\beta}_{p+1}\|$, and assume $B_t = o\left(\sqrt{t}\right)$.

**Theorem 3** (Regret Bound for D-DCCB). *Under the conditions of Theorem 2, with $\tau_{i,t} = \boldsymbol{x}'_{i,t}\boldsymbol{\beta}_{i,t}$, A 3, A3, and A9 in Appendix, we have the dynamic regret of Algorithm 3 is bounded by*

$$\mathcal{O}\left(\frac{N^{1-\omega}T^{1-\alpha}}{\sqrt{\log(N+T)}} + S(N)\log(N)N^{2\omega}T^{1/2+\alpha} + \frac{1}{(1-\rho)^3}\frac{1}{N}\sum_{i=1}^{N}\sum_{t=t_0}^{T}\frac{1}{t-1}B_{t-1}^2\right).$$

Compared to Theorem 2, Theorem 3 includes an additional term that accounts for the non-stationarity of treatment effects. Under A3, this extra term is sublinear in both $T$ and $N$, ensuring that the impact of

---

**Algorithm 3** Discounted Dynamic Causal Contextual Bandits(D-DCCB)

---

**Input:** Burning period $t_0$; Number of agents $N$; Epsilon-Greedy parameter $0 < \epsilon_t < 1$; $0 < \rho < 1$.

**for** Time $t = 1, \cdots, t_0$ **do**

    $a_{i,t} \sim \text{Bernoulli}(0.5), 1 \le i \le N$;

**end for**

**for** Time $t = t_0 + 1, \cdots, T$ **do**

    **for** Arm $a \in \{0, 1\}$ **do**

        **Step 1:** Use equation 3 to estimate $\widehat{\mathbf{M}}_{t-1}(a)$.

    **end for**

    **for** Agent $i = 1, 2, \cdots, N$ **do**

        **Step 2:** Estimate $\widehat{\tau}_{i,s} = \widehat{\mu}_{i,s}(1) - \widehat{\mu}_{i,s}(0)$ for $s = 1, 2, \cdots, t - 1$;

        **Step 3:** Estimate $\widehat{\boldsymbol{\beta}}_{i,t-1} = \left\{ \sum_{s=1}^{t-1} \rho^{t-1-s} \boldsymbol{x}_{i,s} \boldsymbol{x}'_{i,s} \right\}^{-1} \sum_{s=1}^{t-1} \rho^{t-1-s} \boldsymbol{x}_{i,s} \widehat{\tau}_{i,s}$, and estimate the treat-

ment effect as $\widetilde{\tau}_{i,t} = \boldsymbol{x}'_{i,t} \widehat{\boldsymbol{\beta}}_{i,t-1}$

        **Step 4:** Estimate the optimal arm $\widehat{\pi}_{i,t} = \mathbf{1}_{\widetilde{\tau}_{i,t} > 0}$

        **Step 5:** Choose arm $a_{i,t} = \widehat{\pi}_{i,t}$ with probability $(1 - \epsilon_t)$, and $a_{i,t} \sim \text{Bernoulli}(0.5)$ with probability

$\epsilon_t$;

        **Step 6:** Receive reward $r_{i,t}$;

    **end for**

**end for**

---

non-stationarity on the dynamic regret is controlled and does not dominate the total dynamic regret as the number of units or time points increases.

*Remark* 3. We note the linear form of $\tau_{i,t}$ can be easily relaxed to the non-linear case as $\tau_{i,t} = f(\boldsymbol{x})' \boldsymbol{\beta}_{i,t}$, where $f(\cdot)$ is a continuous function. Then the corresponding online estimator for $\tau_{i,t}$ is defined as $\widetilde{\tau}_{i,t} = f(\boldsymbol{x})' \widehat{\boldsymbol{\beta}}_{i,t-1}$.

# 7 Simulation

In this section, we compare our approach, DCMAB and DCCB, with other state-of-the-art methods in literature within the context of two-armed bandit problems using extensive simulation studies. To demonstrate the effectiveness of our proposed method, we consider two data generation settings: Setting 1 evaluates DCMAB in situations of non-stationary Bandits with non-Markovian latent variables and without contextual information, and Setting 2 assesses DCCB in situations of non-stationary Bandits when only partial contextual information can be observed.

**Setting 1: Non-stationary Bandits with Non-Markovian Latent Variables** In this setting, we model the mean reward for arm 0 using a low-rank matrix $\{\mathbb{E}(r_{i,t}(0))\}_{1 \le i \le N, 1 \le t \le T} = 0.8 \mathbf{A}_{N \times r} \mathbf{B}_{r \times T} + 0.2 \mathbf{X}_1 - 0.2 \mathbf{X}_2$, where $r = 25$, and $\mathbf{X}_j = \{\boldsymbol{x}_{i,t,j}\}_{1 \le i \le N, 1 \le t \le T}$, $j = 1, 2$ with $\boldsymbol{x}_{i,t,1} \sim$ i.i.d $\mathcal{N}(0,1)$, and $\boldsymbol{x}_{i,t,2} \sim$ i.i.d $\text{Unif}(-2, 2)$. Each element in $\mathbf{A}_{N \times r}$ follows an i.i.d. normal distribution $\mathcal{N}(0, 1)$, and each row of $\mathbf{B}_{r \times T}$ forms an AR(1) series with a coefficient of 0.9. For each agent $i$, the individual treatment effect $\mathbb{E}\{r_{i,t}(1)\} - \mathbb{E}\{r_{i,t}(0)\}$ is constant over time and i.i.d sampled from a uniform distribution over $(-2, -1) \cup (1, 4)$. The final rewards are generated by $r_{i,t}(a) = \mathbb{E}\{r_{i,t}(a)\} + \epsilon_{i,t}(a)$, with $\epsilon_{i,t}(a)$ i.i.d sampled from $\mathcal{N}(0, 0.1^2)$.

**Setting 2: Non-stationary Bandits with Partially Observable Contextual Information** In this setting, we have contextual information $\boldsymbol{x}_{i,t,1} \sim$ i.i.d $\mathcal{N}(1, 4)$, and $\boldsymbol{x}_{i,t,2} \sim$ i.i.d $\text{Unif}(-2, 2) \in \mathbb{R}^T$. The reward for arm $a = 0$ is generated by $\{r_{i,t}(0)\}_{1 \le i \le N, 1 \le t \le T} = 0.8 \mathbf{X}_1 - 0.8 \mathbf{X}_2 + 3 \mathbf{X}_3$, where $\mathbf{X}_j = \{\boldsymbol{x}_{i,t,j}\}_{1 \le i \le N, 1 \le t \le T}$ for $j \in \{1, 2, 3\}$, and $\{\boldsymbol{x}_{i,t,3}\}_{1 \le t \le T} \sim AR(1) \in \mathbb{R}^T$ is a latent variable. We randomly generate the individualized treatment effect of agent $i$ at time $t$ as $\text{HTE}_{i,t} = x_{i,t,1} + x_{i,t,2}$. As such, the reward for $a = 1$ is defined as $\mathbb{E}\{r_{i,t}(1)\} = \mathbb{E}\{r_{i,t}(0)\} + \text{HTE}_{i,t}$. The final rewards are generated by $r_{i,t}(a) = \mathbb{E}\{r_{i,t}(a)\} + \epsilon_{i,t}(a)$, with $\epsilon_{i,t}(a)$ i.i.d sampled from $\mathcal{N}(0, 0.1^2)$.

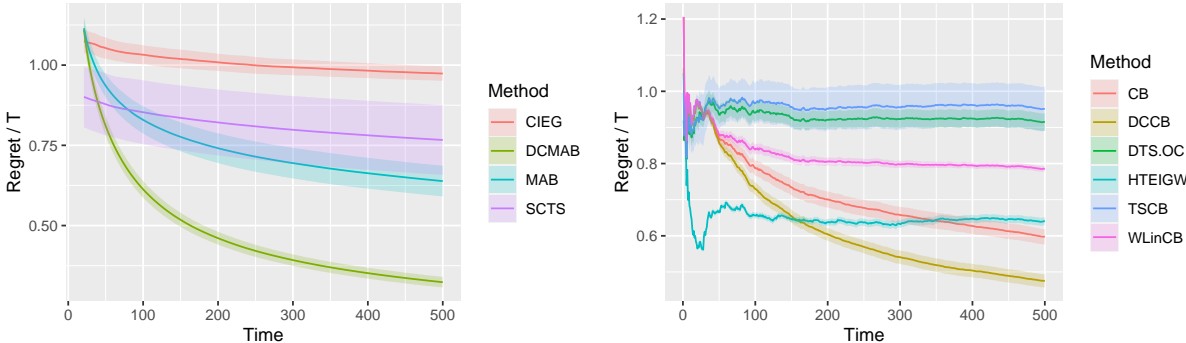

Figure 3: Regret/T in Setting 1 (left) and 2 (right). The confidence band represents one standard deviation. All regrets are obtained based on 100 replications.

In Setting 1, we compare DCMAB with three existing approaches in literature: classical Multi-Armed bandits (MAB), Counterfactual inference for sequential experimental design (CIEG) (Dwivedi et al., 2022), and Synthetically Controlled Thompson Sampling (SCTS) (Farias et al., 2022). In Setting 2, we compare DCCB with four other approaches in literature: classical contextual bandits (CB), Discounted Thompson Sampling with Online Clustering (DTS.OC) (Dzhoha & Rozora, 2023), Heterogeneous Treatment Effect Inverse Gap Weighting (HTEIGW) (Carranza et al., 2023), Thompson Sampling based Contextual Multi-Armed Bandits with Online Scoring and Batch Training (TSCB) (Sawant et al., 2018), and Weighted Linear Contextual Bandits (WLinCB) (Wang et al., 2023).

In this paper, we explored various scenarios with a total of 200 agents and time steps ranging from 1 to 500. We plot the total regret $\mathcal{R}_{N,T}$ divided by $T$ over time in Figure 3. It's evident that our methods (DCMAB and DCCB) consistently achieve lower regret compared to other approaches, demonstrating superior performance whether or not contextual information is available. In Setting 1, shown in the left panel of Figure 3, CIEG utilizes a quasi-k-nearest-neighbor algorithm primarily for counterfactual mean inference. However, it requires a large group of units to remain in the control group, leading to suboptimal regret. SCTS, despite employing synthetic control methods to address nonstationarity and confounding, falls short due to a rigid requirement of a fixed number of units always being assigned to the control group. In Setting 2, depicted in the right panel of Figure 3, DTS.OC relies on clustering for learning reward distributions but fails to quantify unmeasured confounders when clustering structures do not exist, resulting in suboptimal performance. While HTEIGW learns faster at the beginning due to the absence of a burning period, the introduction of a 'safe' period and a restart strategy, as both an innovation and a potential risk of their method, may lead to a suboptimal regret due to challenges in effectively utilizing history information during model fitting. TSCB incorporates causal effect optimization in arm selection but fails to handle non-stationarity effectively, just like classical MAB and CB. In contrast, our approach successfully addresses nonstationarity and confounders simultaneously, achieving the lowest regret in both scenarios.

In our experiments, all simulation results were obtained using standard multi-core parallel computing resources with less than 20GB of memory. The average running time for a single agent is 51.36 seconds for DCMAB and 32.72 seconds for DCCB.(plot). All the code for both the simulation and real data can be found on this anonymous GitHub page. And additional simulation results on k-arm and regret with respect to the change of unit number N can be found in Appendix C.

## 8 Real Data

In this section, we compare the performance of our proposed methods against other approaches using stock market data Athey et al. (2021) (GNU General Public License v3.0). The dataset contains the daily returns of over 2,000 stocks recorded over 3,000 days. For ease of illustration, we selected a subset of 20% of the stocks, totaling $N = 246$, over a span of $T = 200$ continuous days. The original dataset represents a control scenario, to which we've introduced a hypothetical "treatment" effect. We determined these effects

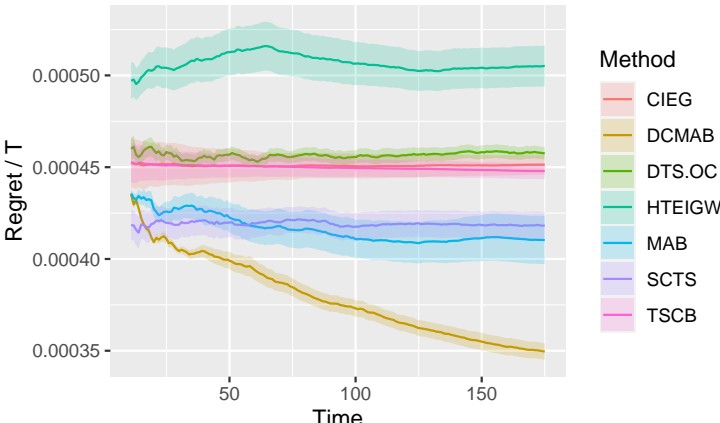

Figure 4: Regret/T v.s. Time in Stock Market. All results are obtained from 100 replications with confidence bands representing one standard deviation.

by randomly drawing from a uniform distribution, $0.1 * \mathrm{Unif}(-q_{0.5}, q_{0.5})$, with $q_\zeta$ denoting the $\zeta$th level quartile of the returns for each stock. As such, the treated reward for a stock on a day is generated by $R_{i,t}(A = 1) = R_{i,t}(A = 0) + \mathrm{ATE}_i + \epsilon_{i,t}$, where the white noise $\epsilon_{i,t}$ follows an independently and identically distributed Normal distribution.

The model comparison results are shown in Figure 4. Notably, DTS.OC, HTEIGW, and TSCB require contextual information. To ensure a comprehensive evaluation, we manually introduce an all-zero covariate as input for these algorithms. (CB and DCCB, being direct contextual versions of MAB and DCMAB, are excluded from this real data analysis.) As The results clearly show that our proposed method achieves the lowest regret over time and performs significantly better than all other baseline approaches. This demonstrates the superiority and flexibility of our approach in handling real data, especially when the structure of non-stationarity and latent confounders is totally unknown.

## 9 Discussion and Open Questions

In this paper, we introduce Dynamic Causal Bandits to address the challenges of non-stationarity and latent variables. Integrating causal inference principles with panel data methodologies, we focus on estimating treatment effects rather than direct reward modeling and achieve sub-linear dynamic regret. The effectiveness of our methods is validated through extensive simulations and a real-world application in the stock market. Our work leaves several questions open for future research. First, establishing a dynamic regret lower bound remains challenging; we conjecture that the dependency on $\sqrt{NT}$ is unavoidable due to the inherent complexity of the problem scaling with these parameters. Moreover, the successful application of our methods indicates promising directions for exploring other panel data approaches within bandit frameworks and extending our findings to other domains such as Markov Decision Processes and Dynamic Treatment Regimes.

**Broader Impact Statement** Our work proposes new policy learning methods within the framework of online contextual bandits, focusing primarily on theoretical advancements in online learning. The proposed methods are general and not restricted to any specific application domain. While our research does not directly address social or ethical implications, the broader impact will depend on how these methods are applied in real-world settings. We encourage responsible and ethical deployment in domains where decision-making affects individuals and society.

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
