# OpenReview forum: "Nonstationary Latent Bandits"
_TMLR — Rejected by TMLR_

### Review · Reviewer_dneq · 2025-05-29

**Summary Of Contributions:**

This paper considers the question of non-stationary multi-agent MAB with latent structure on the rewards. For this setting, they use matrix completion methods on the matrix of reward (for each arm) w.r.t. time x agents, thereby reducing the regret analysis to estimation error bounds for matrix completion. They show sublinear regret bounds for this setting and also extend to a linear contextual model. They also show superior performance of their methods on simulations of real data.

**Audience:**

Yes

**Broader Impact Concerns:**

N/A.

**Claims And Evidence:**

Yes

**Requested Changes:**

## Questions/Notes
* The statement of Theorem 1 is a bit opaque. First, the assumption on $S(N)$ seems to restrict the non-stationarity heavily as the rank of the worst-case reward matrix cannot change in time (which already seems to me to be close to assuming stationarity anyways). And, how does this assumption compare to the also used Assumption 3 here? The writing is a bit confusing since in Section 6, you say you extend DCCB algorithm to "handle scenarios involving non-stationary..." but in Section 5 you also seem to assume the full non-stationary model?
* Furthermore, how should one set $\omega,\alpha$ to get the best regret bounds or bounds which compare favorably to previous works? Although the authors emphasize there are no known works on addressing all three of (1) non-stationarity, (2) latent structure, and (3) multi-agents, it would still be necessary to do a more thorough comparison with the previous art bounds in more limited settings (addressing either two of (1)-(3) or operating under more limited assumptions). Such a comparison is missing in the submission, which makes it hard to place the results in the context of the literature.
* Following this thread of thought, do the results here recover single-agent non-stationary bandit dynamic regret bounds in rested settings?
* In fact, comparisons with previous works is made doubly difficult by the fact that all the regret bounds seem to assume $T\leq N$ which for me makes it hard to compare to the single-agent setting.
* For Theorem 3, is $\rho$ being set in terms of $B_t$ or its bound $\sqrt{t}$? This should be made more clear.
* Assumption A3 is also a bit restrictive in only allowing a certain amount of non-stationarity. There could be more discussion on why such an assumption is required (even if it is for the sake of analysis) and to what extend the algorithm requires knowledge of $B_t$ or its bound.

## Writing Notes
* Some citations are missing parentheses: e.g., in Example 1, "Athey et al. (2021)", or the same citation later in the first sentence of Section 8.
* Many paragraph titles are missing colons or periods, e.g. "Challenge 1: Dynamic Nature of Rewards".
* The labels in Figure 2 are small, and a bit hard to read.
* In some places, there is inconsistent usage of brackets vs. parentheses in expectations, e.g., equation (1).
* Redundant naming "Equation equation 3" in Step 1 of Algorithm 1
* It'd be better to write, e.g., "A 1" as "Assumption 1" to make it more transparent to the reader.
* In Theorem 1, assumption 3 is invokved before it is introduced. Also, I don't know why you write "(assumption) AA1" when you already say "A1". Furthermore, I believe you mean to say "$S(N)$ remains constant as $t$ increases not $T$. Perhaps the notation $S(N)$ should also involve $t$?
* "variation budget" should be capitalized in A3.
* The trace operator appearing in proofs (e.g., p. 19 of supplement) should be in normalfont and not italicized.

**Strengths And Weaknesses:**

The work addresses an important combination of models/settings using a (at least to my knowledge) novel usage of matrix completion, and also has compelling simulation results. However, I find the theoretical results to be a bit opaque and are owed more clarification on how the regret upper bounds compare to those of previous works. See my more specific questions below. There are also many typos and the writing can be confusing at times, which for me is primarily due to the lack of clarity of how to contextualize the contribution in the broader literature.

As such, I believe a round of revision or discussion is required to get a better sense of novelty and scope of results.

---

> ### Author Response · Authors · 2025-06-22
> **Point-by-point Response to Reviewer dneq Part I**
>
> Dear Reviewer dneq,
>
> Thank you very much for your comments. The following are our point-by-point responses.
>
> * **In response to Questions/Notes bullet 1:**
>
> First, we thank the reviewer for their question regarding our assumption on the order of $S(N)$. This assumption is a theoretical requirement originated from matrix completion. Specifically, the order of the $N \times t$ reward mean matrix $M_t(a)$ should generally remain constant in $t$ to ensure that it is estimable via low-rank matrix completion techniques.
>
> We emphasize that this is not "close to assuming stationarity", as stationarity is just a strong special case. Instead, our assumption allows $M_t(a)$ to exhibit a finite-dimensional structure that may capture periodic or other latent patterns, without requiring us to pre-specify when these patterns occur, how they evolve, or any parametric form. All structural patterns are learned directly from the data through matrix completion, which is substantially more flexible than prior work that allows for nonstationarity but imposes specific structural assumptions, such as Bogunovic et al. 2016, Cai et al. 2021, Deng et al. 2022.  This assumption is widely used in applications such as matrix completion (Athey et al. 2021) and synthetic control (Abadie et al. 2010), and is compatible with data-generating processes like aggregated data (Shi et al. 2021, Abadie et al. 2010, Dube & Zipperer 2015, Bayat et al. 2020, Chen
> et al. 2021), tensor factor models (Agarwal et al. (2021), Bayat et al. (2020)) and autoregressive (AR) models (Ben-Michael et al. (2022), Abadie (2021)). For example, in synthetic control, this assumption implies that the potential outcome of one treated unit can be well approximated by the weighted average of some control units. This low-rank structure effectively restricts the temporal complexity of reward variation, while still accommodating flexible non-stationarity. Specifically, it rules out adversarial or arbitrarily fast-changing reward processes - precisely the kinds of behavior that lead to linear regret in more general dynamic settings.
>
> Moreover, since $S(N)$ is only a theoretical construct, it does not need to be specified in practice. The matrix completion algorithm implicitly handles this aspect, and $S(N)$ can be taken to be a sufficiently large constant if needed. Therefore, we believe this condition does not impose any substantial restriction in real-world applications.
>
> Regarding the relationship with Assumption 3, we clarify that the two assumptions are entirely orthogonal. The quantity $S(N)$ characterizes the rank of the reward mean matrix $M_t(a)$, which is related to the learnability of its low-rank structure. In contrast, Assumption 3 controls the temporal variation of the parameter sequence $\boldsymbol{\beta}_t$, requiring that it does not vary faster than $o(\sqrt{t})$. This assumption is introduced specifically in our extension to the non-stationary setting, where $\boldsymbol{\beta}_t$ is allowed to evolve over time.

---

> ### Author Response · Authors · 2025-06-22
> **Point-by-point Response to Reviewer dneq Part II**
>
> **In response to Questions/Notes bullet 2 on $\omega$ and $\alpha$:** In the regret upper bound presented in Theorem 1, if we omit lower-order terms (such as $\sqrt{\log(N+T)}$, $\log N$, and $S(N)$) which have a negligible impact on the leading order, the regret simplifies to an upper bound of approximately $\mathcal{O}(N^{2/3}T^{3/4})$ when $\omega = 1/3$ and $\alpha = 1/4$. This rate represents the minimal achievable order under our formulation.
>
> **In response to Questions/Notes bullet 2 and 3 on the regret bound:** Regarding the reviewer's question about recovering existing regret bounds that address at least two of the three challenges, we clarify that:
>
> 1. Our algorithm is specifically designed for the multi-agent setting with non-stationarity and latent variables. It leverages cross-agent structure through low-rank assumptions, which are not present in single-agent formulations. Therefore, it is not directly applicable to single-agent problems, nor is it intended to recover existing regret bounds for such settings as special cases.
> 2. Importantly, our incorporation of non-stationarity and latent structure does not impose stronger assumptions; rather, it relaxes common assumptions such as stationarity and full observability. Consequently, it is natural that our regret bounds, which hold under these more general conditions, are not tighter than those derived for more restrictive settings. For instance,  Cheung et al. (2022) derived a regret bound of $\widetilde{O}\left(d^{2/3} B_T T^{2/3}\right)$, Deng et al. (2022) obtained a bound of $\widetilde{O}\left(\dot{\gamma}_T^{7/8} B_T T^{3/4}\right)$, where $\dot{\gamma}_T$ denotes the maximum information gain, and Russac et al. (2019) achieved a bound of $O\left(d^{2/3} B_T^{1/3} T^{2/3}\right)$. Within in the literature of stationary contextual bandits with latent variables, Wang et al. (2016) derived a regret bound of $O(\sqrt{T} \ln T)$ by directly introducing latent features into the linear bandits. Our algorithm is not designed to recover these bounds in simpler scenarios, as it targets a different goal: achieving sublinear dynamic regret under the joint presence of latent heterogeneity and non-stationarity. While our regret bound may be looser in $T$ compared to those obtained in more restricted settings, it holds under significantly weaker and more realistic assumptions. This reflects the standard trade-off in theoretical analysis: stronger generality often necessitates relaxed performance guarantees.
> 3. We have a comparative discussion at the end of Section 5 in our original manuscript to better contextualize our results with respect to prior work. For the reviewer’s convenience, we have attached it here here:  While Sawant et al. (2018) pioneered the integration of single-stage treatment effect estimators in an online bandit setting, they did not provide theoretical analysis. On the other hand, Carranza et al. (2023) could only achieve linear regret in $T$. In comparison with the literature on MAMAB with heterogeneous feedback, the only algorithm that allows heterogeneous feedback within clusters attains a regret bound of $ \mathcal{O}\left( N\sqrt{T\log(T)} \right)$ (Wang et al. 2021), which is suboptimal in $N$. Although our regret bound grows faster with $T$, our approach accounts for the non-stationarity of rewards, a factor not considered by Wang et al. (2021), who assume stationary rewards over time. Overall, our approach attains a dynamic regret bound that is sublinear in both $N$ and $T$, resulting in a more favorable regret performance.

---

> ### Author Response · Authors · 2025-06-22
> **Point-by-point Response to Reviewer dneq Part III**
>
> **In response to Questions/Notes bullet 4:**
>
> We appreciate the reviewer's interest in comparing our results to the single-agent setting. We would like to clarify that our algorithm is explicitly designed for the multi-agent case, where the latent structure across agents is essential for information sharing. This structure is both critical for identifiability under non-stationarity and a key part of our design motivation. Our regret bounds therefore do not directly reduce to those in the single-agent literature, nor is our algorithm applicable to single-agent problems. In fact, applying our method to a single-agent setting would remove the possibility of estimating latent structure.
>
> Regarding the assumption of $T < N$, this setting is motivated by practical scenarios where each agent is observed for a relatively short duration, but a large number of agents are available. Examples include mobile health interventions, online education platforms, or early-stage clinical trials, where the number of individuals exceeds the observation window for each. This assumption is also consistent with prior work in related domains (Jain & Pal 2022, Pal et al. 2023, Farias et al. 2022). In these works, the low-rank structure helps compensate for the short time horizon by leveraging cross-sectional information.
>
> We hope this clarifies the distinct modeling assumptions and technical contributions of our work. A direct comparison to single-agent results would thus be misleading, and we instead provide comparison to the closest multi-agent settings in Section 5 (highlighted in blue).

---

> ### Author Response · Authors · 2025-06-22
> **Point-by-point Response to Reviewer dneq Part IV**
>
> **In response to Questions/Notes bullet 5:**
>
> We thank the reviewer for highlighting the need to clarify the role of $\rho$ in our regret bound.
>
> In Theorem 3, the regret bound is explicitly expressed as a function of $\rho$, which controls the weight assigned to historical information during estimation. A larger $\rho$ places more emphasis on past observations, effectively reducing the discounting of older data. This introduces a trade-off between bias and variance in estimating time-varying treatment effects.
>
> We agree that this dependence deserves clearer exposition in the main text. While we do not derive a closed-form optimal choice of $\rho$ due to the complexity of the non-stationary and multi-agent setting, it is intuitive that the optimal $\rho$ would depend on both the time horizon $T$ and the underlying variation budget $B_T$, similar to $\rho=1-\max \\{1 / T, \sqrt{ B_T /(d T) }\\}$ as derived in Wang et al. (2023) for simpler settings. In practice, $\rho$ can be treated as a tunable hyperparameter, selected via validation or based on domain expertise.
>
> **In response to Questions/Notes bullet 6:** Assumption 3 is used to quantify the regret upper bound in our extension where $\boldsymbol{\beta}_t$ is non-stationary over time. Specifically, in Equation (20) of the Supplementary Material, we decompose the regret into two components via the triangle inequality: the first line $\eta_1$ (omitting subscripts $i$ and $t-1$ for simplicity), which captures the non-stationarity-induced error due to temporal changes in $\boldsymbol{\beta}$ from round $s$ to $t$, and the second line $\eta_2$, which accounts for the statistical estimation error arising from finite samples. Assumption 3 directly controls the term $\eta_1$ by limiting the rate at which $\boldsymbol{\beta}_t$ can vary over time. This type of assumption is commonly used in the literature on non-stationary bandits with time-varying parameters, such as in Wang et al. (2023).
>
> **In response to Writing Notes:** We sincerely appreciate the reviewer’s careful reading and thoughtful suggestions regarding the writing, including corrections of typographical errors. We will carefully incorporate each suggestion in the final version of the paper to improve clarity and readability for the broader audience.

---

> ### Author Response · Authors · 2025-06-22
> **References**
>
> 1. Abadie, A. (2021), ‘Using synthetic controls: Feasibility, data requirements, and methodological aspects’, Journal of economic literature 59(2), 391–425.
> 2. Abadie, A., Diamond, A. & Hainmueller, J. (2010), ‘Synthetic control methods for comparative case studies: Estimating the effect of california’s tobacco control program’, Journal of the American statistical Association 105(490), 493–505.
> 3. Agarwal, A., Shah, D. & Shen, D. (2021), ‘Synthetic interventions’, arXiv preprint arXiv:2006.07691.
> 4. Athey, S., Bayati, M., Doudchenko, N., Imbens, G. & Khosravi, K. (2021), ‘Matrix completion methods for causal panel data models’, Journal of the American Statistical Association 116(536), 1716–1730.
> 5. Bayat, N., Morrin, C., Wang, Y. & Misra, V. (2020), ‘Synthetic control, synthetic interventions, and covid-19 spread: Exploring the impact of lockdown measures and herd immunity’, arXiv preprint arXiv:2009.09987.
> 6. Ben-Michael, E., Feller, A. & Rothstein, J. (2022), ‘Synthetic controls with staggered adoption’, Journal of the Royal Statistical Society Series B: Statistical Methodology 84(2), 351–381.
> 7. Bogunovic, I., Scarlett, J. & Cevher, V. (2016), Time-varying gaussian process bandit optimization, in ‘Artificial Intelligence and Statistics’, PMLR, pp. 314–323.
> 8. Cai, H., Cen, Z., Leng, L. & Song, R. (2021), ‘Periodic-gp: Learning periodic world with gaussian process bandits’, arXiv preprint arXiv:2105.14422.
> 9. Carranza, A. G., Krishnamurthy, S. K. & Athey, S. (2023), Flexible and efficient contextual bandits with heterogeneous treatment effect oracles, in ‘International Conference on Artificial Intelligence and Statistics’, PMLR, pp. 7190–7212.
> 10. Chen, Q., Golrezaei, N. & Bouneffouf, D. (2021), ‘Dynamic bandits with temporal structure’, Available at SSRN 3887608.
> 11. Cheung, W. C., Simchi-Levi, D. & Zhu, R. (2022), ‘Hedging the drift: Learning to optimize under nonstationarity’, Management Science 68(3), 1696–1713.
> 12. Deng, Y., Zhou, X., Kim, B., Tewari, A., Gupta, A. & Shroff, N. (2022), Weighted gaussian process bandits for non-stationary environments, in ‘International Conference on Artificial Intelligence and Statistics’, PMLR, pp. 6909–6932.
> 13. Dube, A. & Zipperer, B. (2015), ‘Pooling multiple case studies using synthetic controls: An application to minimum wage policies’.
> 14. Farias, V., Moallemi, C., Peng, T. & Zheng, A. (2022), ‘Synthetically controlled bandits’, arXiv preprint arXiv:2202.07079.
> 15. Jain, P. & Pal, S. (2022), ‘Online low rank matrix completion’, arXiv preprint arXiv:2209.03997.
> 16. Pal, S., Suggala, A. S., Shanmugam, K. & Jain, P. (2023), ‘Optimal algorithms for latent bandits with cluster structure’, arXiv preprint arXiv:2301.07040.
> 17. Russac, Y., Vernade, C. & Capp´e, O. (2019), ‘Weighted linear bandits for non-stationary environments’, Advances in Neural Information Processing Systems 32.
> 18. Sawant, N., Namballa, C. B., Sadagopan, N. & Nassif, H. (2018), ‘Contextual multi-armed bandits for causal marketing’, arXiv preprint arXiv:1810.01859.
> 19. Shi, C., Sridhar, D., Misra, V. & Blei, D. M. (2021), ‘On the assumptions of synthetic control methods’, arXiv preprint arXiv:2112.05671.
> 20. Wang, H., Wu, Q. & Wang, H. (2016), Learning hidden features for contextual bandits, in ‘Proceedings of the 25th ACM international on conference on information and knowledge management’, pp. 1633–1642.
> 21. Wang, J., Zhao, P. & Zhou, Z.-H. (2023), Revisiting weighted strategy for non-stationary parametric bandits, in ‘International Conference on Artificial Intelligence and Statistics’, PMLR, pp. 7913–7942.
> 22. Wang, Z., Zhang, C., Singh, M. K., Riek, L. & Chaudhuri, K. (2021), Multitask bandit learning through heterogeneous feedback aggregation, in ‘International Conference on Artificial Intelligence and Statistics’, PMLR, pp. 1531–1539.

---

### Review · Reviewer_DmAR · 2025-06-02

**Summary Of Contributions:**

This work studies multi-agent non-stationary bandits. The main idea is to rely on matrix factorization and low-rank assumptions to estimate the time-varying reward functions for all agents and drive the decision making using epsilon-greedy exploration. The paper presents both the standard MAB and the linear bandit version. For both algorithms, they prove regret bounds and they run simulations on synthetic and real data.

I have a number of questions and concerns that need to be addressed in rebuttal / discussion phases before I can make a final decision.

**Audience:**

Yes

**Claims And Evidence:**

No

**Requested Changes:**

### Major Requested Changes

- **Clarify regret definition**
  - Explicitly state that the paper uses *dynamic regret*.
  - Explain how the constant rank assumption constrains reward variation over time.
  - Address why trivial linear regret under arbitrary changes does not apply here.

- **Theorem 1 assumptions**
  - Clarify the need for \( T < N \) and discuss feasibility when \( T \gg N \).

- **Dimension and contextual setting in DCCB**
  - Explain why regret bounds do not depend on problem dimension.
  - Clearly describe the contextual setting and its role relative to the paper’s main contributions.

- **Computational complexity**
  - Provide an explicit discussion of the algorithm’s computational complexity.

### Minor changes
(please see strengths and weaknesses)

**Strengths And Weaknesses:**

### Strengths:
* The paper makes an attempt at bridging causal inference and bandits
* The proposed solution is very general and applies across a wide range of assumptions on the bandit setting (though they are not always very clear to me in the current draft)

### Major remarks:
* Regret definition and relationship with the rank assumptions. The regret is defined on the last line of Section 3 as the sum over N agents of the (dynamic) regrets. I emphasize “dynamic” because this notion is a bit too silent in the paper and it is crucial in non-stationary bandits. Here you want to compare at each round with the optimal action in round t. It is easy to construct problems in general for which this is trivially linear for any reasonable algorithm, typically if the reward function changes every round. In your paper, you constrain the rank of the reward function matrix M to be constant, but it is reallly unclear to me how it translates into constraints on the actual variations of the reward function. Typically, for your problem to be feasible, I would expect some sort of variational budget to hold, like \sum_t \|\mu_t(a) - \mu_{t+1}(a)\| \leq B_T (not sure how it should be done across agents). Can you please explain why the failure case I describe above does not apply to your setting?
* Statement of theorem 1: Is it not a problem that T<N? Is that not a strong assumption? Typically I would like T to grow much larger than the number of agents…
* DCCB and contextual setting: why does the dimension of the problem play no role in the regret bound (theorem 2 and 3)? standard regret bounds for linear bandits depend on this dimension, and on whether the contexts (feature vectors x_{i,t} are fixed or changing arbitrarily). This setting would deserve to be better described. I am also not sure it fully belongs to this paper because the main contribution seems to be orthogonal to the problem of having contexts or not.
* Could you please explicitly discuss the computational complexity of your algorithm?


Minor remarks:
* Under assumption 1: citations are a bit all over the place and do not all justify clearly the statement that “MAB often implicitly make A1”. Typically Fevotte et al 2009 is not a bandit paper. Please check that citations are appropriate everywhere.
* Figure 1 is not easy to read. The distinction between 1a/1b is *only* the index t for \mu (same for 1c/1d). Could this be better highlighted? Otherwise I really don’t see the point of the figure. Maybe just show 1a and 1c and highlight a (t) index for \mu?
* Notation consistency for A_{i,t} vs a_{i,t}: in assumption A2, A_{i,t} is used, while a_{i,t} is introduced in section 3. Is there a difference?
* below A2: “this is evident in semi-parametric bandits [citations]…”: What is this sentence supposed to justify? You want to justify that “A2 does not exclude the existence of latent variables” and I don’t think citations can help you doing so. I am fine with the sentence just after that actually justifies the claim.
* Eq 1 and above: Can you please clarify somewhere the difference between E{} and E()? I think I agree with the equalities claimed in these blocks but I don’t think the notation {}-> () helps to understand. I think it’s rather confusing.
* page 6: “Equation equation 3”

---

> ### Author Response · Authors · 2025-06-22
> **Point-by-point Response to Reviewer DmAR Part I**
>
> Hi Reviewer DmAR,
>
>
> Thank you very much for your comments. The following are our point-by-point responses.
>
> **Explicitly state that the paper uses dynamic regret:**
>
> We thank the reviewer for raising these  insightful points. We confirm that our regret formulation corresponds to dynamic regret, as it compares the performance of the algorithm at each time point to the optimal action at that same time. This is a natural choice in our setting, where treatment effects and covariates may vary over time. We have clarified this in the revised manuscript and highlighted the relevant changes in blue to improve clarity.
>
>
> We also agree that in non-stationary bandit settings, it is crucial to specify how assumptions constrain reward variation to avoid cases with linear regret. In our work, we impose a low-rank assumption on the reward matrix, formalized as $ S(N) \triangleq  \max_{a \in A} \{ \sigma_{a}^2 S_a(N,t) \}  $. This assumption, motivated by the causal inference literature, implies that for each action $a$, the reward trajectories across time and units lie approximately in a shared low-dimensional subspace.  It is widely used in applications such as matrix completion (Athey et al. 2021) and synthetic control (Abadie et al. 2010), and is compatible with data-generating
> processes like aggregated data (Shi et al. 2021, Abadie et al. 2010, Dube & Zipperer 2015, Bayat et al. 2020), tensor factor models (Agarwal et al. 2021, Bayat et al. 2020) and autoregressive (AR) models (Ben-Michael et al. 2022, Abadie 2021). For example, in synthetic control, this assumption implies that the potential outcome of one treated unit can be well approximated by the weighted average of some control units. This low - rank structure effectively restricts the temporal complexity of reward variation while still accommodating flexible non-stationarity. Crucially, it rules out adversarial or arbitrarily fast-changing reward processes—precisely the kinds of behavior that lead to linear regret in more general dynamic settings.
>
>
> We view this structural assumption as complementary to classical total variation budget assumptions (e.g., $\sum_t |\mu_t(a) - \mu_{t+1}(a)| \le B_{T}$). While these assumptions constrain reward smoothness directly, our formulation offers an alternative and interpretable way to regulate non-stationarity through latent structure. This perspective enables us to connect with the causal inference literature and is a key innovation of our approach.
>
>
> Finally, in the extension section of the paper, where we consider time-varying treatment effects, we explicitly adopt a variation budget assumption on the treatment effect trajectories (Assumption A3). This further demonstrates our attention to temporal complexity and model realism.
>
>
> **Theorem 1 assumptions:** We thank the reviewer for raising this important point. Our theoretical results, particularly Theorem 1, are derived under the condition that the number of time periods $T$ is less than the number of agents $N$, i.e., $T < N$. This setting is motivated by practical scenarios where each agent is observed for a relatively short duration, but a large number of agents are available. Examples include mobile health interventions, online education platforms, or early-stage clinical trials - contexts where the number of individuals exceeds the observation window for each. This assumption is also consistent with prior work in related domains (Jain & Pal 2022, Pal et al. 2023, Farias et al. 2022). In these works, the low-rank structure helps compensate for the short time horizon by leveraging cross-sectional information. We agree that understanding the behavior of the algorithm in the regime $T \gg N$ is also important, especially in long-running systems with limited sample sizes. While our current theory does not explicitly address this regime, the algorithm remains applicable, and we view extending the analysis to the $T \gg N$ case as a valuable direction for future work.
>
> **Dimension and contextual setting in DCCB:**  We thank the reviewer for raising this thoughtful question regarding the role of context dimensionality in our regret analysis. In our setting, the feature dimension $d$ is treated as fixed and does appear explicitly in the constants of the regret bounds, as detailed in the appendix. As the reviewer correctly points out, the core theoretical contribution of our work is largely orthogonal to whether context features are present. Thus, to maintain clarity and focus in the main text, we treat the feature dimension $d$ as fixed and emphasize the scaling of regret with respect to the number of agents $N$ and the time horizon $T$. This allows us to highlight the core novelty of our work—handling latent, non-stationary treatment effects using low-rank structure—without conflating it with the well-studied complexity introduced by high-dimensional covariates. We have revised the manuscript to clarify the contextual assumptions and highlighted them in blue.

---

> > ### Author Response · Authors · 2025-06-22
> > **Point-by-point Response to Reviewer DmAR Part II**
> >
> > **Computational complexity:**
> >
> > We thank the reviewer for this insightful comments. Below, we provide an explicit discussion of the computational complexity of our algorithms.
> >
> > Our algorithm involves a matrix completion step in each round $t$, which dominates the per-round computational cost. Specifically, The matrix completion step operates on an $N \times t$ reward matrix and involves iterative singular value shrinkage. Denoting $K_t$ as the number of iterations until convergence at round $t$, the per-round complexity of this step is $\mathcal{O}(K_t N t^2)$.
> >
> > For Dynamic Causal Multi-Armed Bandits (DCMAB), each agent uses sample mean, leading to a per-round cost of $\mathcal{O}(N t)$, therefore, the total computational complexity across $T$ rounds is  $\sum_{t=1}^{T} \mathcal{O}(K_t N t^2)$.  If $K_t \approx K$ is treated as a constant,  the total computational complexity across $T$ rounds is  $\mathcal{O}(K N T^3)$.
> >
> >
> >
> > For contextual setting (DCCB) and Discounted Dynamic Causal Contextual Bandits (D-DCCB), each agent fits a linear regression model with $t$ samples and $d$ feature dimensions, leading to a per-round cost of $\mathcal{O}(N t d^3)$. Therefore, the total computational complexity across $T$ rounds is  $\sum_{t=1}^{T} \mathcal{O}(K_t N t^2+N t d^3)$. If $K_t \approx K$ is treated as a constant,  the total computational complexity across $T$ rounds is  $\mathcal{O}(K N T^3 + NT^2d^3)$.
> >
> > As $T$ increases, the time required for matrix completion grows, but this can be mitigated with a warm-start initialization. In round $t+1$, matrix completion on $N\times (t+1)$ matrix $R_{t+1}(a)$ can start from the previous result $M_t(a)$ , as the first $t$ columns are likely similar, which significantly reduces computational time. In our implementation, we initialize round $t+1$ with $[M_t(a),R_{t+1}(a)^{(t+1)}]$, where $R_{t+1}(a)^{(t+1)}$ denotes the $(t+1)$-th column of matrix $R_{t+1}(a)$, a vector of length $N$. This approach ensures efficient updates while maintaining accuracy.
> >
> >
> > In Figure 3 in the appendix, we compare the computational time of all methods implemented in the simulation. As shown, both DCCB and DCMAB exhibit reasonable computational times, comparable to existing approaches that address non-stationarity or latent variables.
> >
> >
> > **Minor Remarks 1:** Thank you for pointing out the citation to Fevotte et al. (2009). We agree that it is not relevant in this context and have removed it in the final version of the paper. We have also conducted an additional round of careful review to ensure that all remaining citations are appropriate and correctly placed.
> >
> > **Minor Remarks 2:** Thank you for your question regarding Figure 2 (we assume this was a typo, as your comments referred to Figure 1). Let us clarify the differences between subplots 1a–1d and explain their significance. Our paper addresses two core challenges simultaneously: non-stationarity and latent variables. The transition from 1a to 1b (and similarly from 1c to 1d) is intended to illustrate the effect of non-stationarity. As you correctly pointed out, subplots 1a–1b demonstrate that the action-reward relationship, denoted by $\mu_t(A_{i,t})$, is no longer stationary and may vary arbitrarily over time. This introduces substantial complexity and would invalidate classical bandit algorithms that rely on cumulative learning with stationary reward functions. We sincerely appreciate your suggestion to highlight the time-varying subscript, and we agree that doing so would improve clarity. We have revised the figure captions to explicitly differentiate between $\mu(A_{i,t})$ and $\mu_t(A_{i,t})$, using appropriate notation and parenthetical explanations.
> >
> >
> > **Minor Remarks 3:** Thank you for pointing this out. Throughout the paper, we use capital letters (e.g., $S_{i,t}$, $A_{i,t}$, $R_{i,t}$) to denote random variables and lowercase letters (e.g., $s_{i,t}$, $a_{i,t}$, $r_{i,t}$) to denote their realizations. We have included a clarifying sentence in the revised version to make this convention explicit.
> >
> >
> > **Minor Remarks 4:** Below Assumption A2, we originally wrote "this is evident in semi-parametric bandits [citations]..." to indicate that the condition stated in A2, i.e., the conditional independence between $R(a)$ and $A$, is not necessarily violated in the presence of latent variables. The cited papers support this point, as they all involve latent variables while also adopting an assumption similar to A2. However, we acknowledge that the original phrasing may be unclear. To improve clarity, we have revised the sentence to "This assumption is widely adopted in many bandit works that involve latent variables [citations]...". We hope this change more accurately conveys our intent.

---

> > > ### Author Response · Authors · 2025-06-22
> > > **Point-by-point Response to Reviewer DmAR Part III**
> > >
> > > **Minor Remarks 5:** Thank you for your thoughtful comment regarding the use of different expectation notations (e.g., $\mathbb{E}{}$ vs. $\mathbb{E}()$). To clarify, our convention throughout the paper follows a nesting order for delimiters: parentheses () are used for the innermost expressions, braces {} for intermediate levels, and brackets [] for the outermost layer. If a fourth level of nesting is required, we cycle back to parentheses (). This approach is commonly adopted to improve clarity when multiple layers of functions or expectations are involved. That said, we recognize that this convention may not be universally familiar and could cause confusion. To address this, we will include a brief note in the notation section to explicitly state this delimiter convention. We hope this resolves any ambiguity and improves the reader's experience.
> > >
> > > **Minor Remarks 6:** page 6: ``Equation equation 3" -- Thanks for pointing it out. We will fix this typo in the final version of our paper.
> > >
> > >
> > > Thank you again for your careful reading and thoughtful suggestions. We hope our responses have effectively clarified the points you raised.
> > >
> > > Best regards,
> > >
> > > Authors of Paper4805

---

> ### Author Response · Authors · 2025-06-22
> **References**
>
> 1. Abadie, A. (2021), ‘Using synthetic controls: Feasibility, data requirements, and methodological aspects’, Journal of economic literature 59(2), 391–425.
>
> 2. Abadie, A., Diamond, A. & Hainmueller, J. (2010), ‘Synthetic control methods for comparative case studies: Estimating the effect of california’s tobacco control program’, Journal of the American statistical Association 105(490), 493–505.
>
> 3. Agarwal, A., Shah, D. & Shen, D. (2021), ‘Synthetic interventions’, arXiv preprint arXiv:2006.07691 .
>
> 4. Athey, S., Bayati, M., Doudchenko, N., Imbens, G. & Khosravi, K. (2021), ‘Matrix completion methods for causal panel data models’, Journal of the American Statistical Association 116(536), 1716–1730.
>
> 5. Bayat, N., Morrin, C., Wang, Y. & Misra, V. (2020), ‘Synthetic control, synthetic interventions, and covid-19 spread: Exploring the impact of lockdown measures and herd immunity’, arXiv preprint arXiv:2009.09987.
>
> 6. Ben-Michael, E., Feller, A. & Rothstein, J. (2022), ‘Synthetic controls with staggered adoption’, Journal of the Royal Statistical Society Series B: Statistical Methodology 84(2), 351–381.
>
> 7. Dube, A. & Zipperer, B. (2015), ‘Pooling multiple case studies using synthetic controls: An application to minimum wage policies’.
>
> 8. Farias, V., Moallemi, C., Peng, T. & Zheng, A. (2022), ‘Synthetically controlled bandits’, arXiv preprint arXiv:2202.07079 .
>
> 9. Jain, P. & Pal, S. (2022), ‘Online low rank matrix completion’, arXiv preprint arXiv:2209.03997 .
>
> 10. Pal, S., Suggala, A. S., Shanmugam, K. & Jain, P. (2023), ‘Optimal algorithms for latent bandits with cluster structure’, arXiv preprint arXiv:2301.07040 .
>
> 11. Shi, C., Sridhar, D., Misra, V. & Blei, D. M. (2021), ‘On the assumptions of synthetic control methods’, arXiv preprint arXiv:2112.05671 .

---

### Review · Reviewer_8S5e · 2025-06-07

**Summary Of Contributions:**

The paper proposes a bandit algorithm designed to infer the difference between arm means in a nonstationary setting, where arm rewards may change over time. It analyzes the algorithm’s regret and supports the findings with numerical experiments.

**Audience:**

Yes

**Claims And Evidence:**

No

**Requested Changes:**

1. In Section 3, the authors claim that the goal of the bandit algorithm is to estimate the reward of each arm and select the one with the highest empirical estimate. This is inaccurate. Not all bandit algorithms aim to maximize reward, and the described approach represents only one class of algorithms. This statement should be corrected.

2. There is a conceptual inconsistency: Section 3 frames the goal as learning an optimal policy to minimize cumulative regret, whereas Section 4.2 states that the objective is to optimize the causal effect rather than direct reward. The relationship between these goals—cumulative regret, optimal policy, and causal effect—must be clarified.

3. The authors assume a linear reward-generating mechanism with respect to latent variables, enabling the use of matrix completion. However, this assumption is not clearly stated in the main text or appendix. While Assumptions 5 and 6 in Appendix D may imply it, this should be made explicit in the main body, especially since matrix completion is only applicable under strong and narrow conditions in causal inference.

4. The paper assumes that treatment effects are stationary over time, which significantly simplifies the nonstationary setting and underlies the regret analysis (under Assumptions 1–7 in Appendix D). There are two issues here: (a) these assumptions are not stated in the theorem statements, and (b) the abstract and introduction fail to acknowledge these strong assumptions—namely, treatment effect stationarity and linearity with respect to context.

**Summary**: The analysis presented does not align with the claims made in the abstract and introduction. The proposed algorithm operates under a highly restricted set of assumptions, many of which are not clearly disclosed. This discrepancy could mislead readers and should be addressed through substantial revision.

**Strengths And Weaknesses:**

**Strengths**: The paper addresses the challenging nonstationary setting and proposes an algorithm tailored to infer differences in arm means. The authors support their approach with a regret analysis, offering insight into the algorithm’s convergence rate.

**Weaknesses**: The paper’s writing lacks clarity, with several inaccurate claims and missing key assumptions in the main text. The modeling assumptions are also quite strong, which limits the novelty of the contribution. I recommend revising the introduction and abstract for clarity, and correcting the inaccuracies to avoid misleading readers.

---

> ### Author Response · Authors · 2025-06-22
> **Point-by-point Response to Reviewer 8S5e Part I**
>
> Hi Reviewer 8S5e,
>
> Thank you very much for your comments. The following are our point-by-point responses.
>
> **Requested Change 1:**  We thank the reviewer for the comment.  While we agree that bandit algorithms encompass a variety of approaches, we maintain that in the classical formulation, the primary objective is to learn a policy that maximizes cumulative rewards. To avoid overgeneralization, we have revised the sentence to: "The goal of classical bandit algorithms is to learn a policy that maximizes cumulative rewards." This revision more accurately reflects the standard setting and acknowledges the broader spectrum of bandit variants.
>
>
> **Requested Change 2:** We thank the reviewer for highlighting this point and appreciate the opportunity to clarify. The goal throughout our paper is indeed to maximize cumulative reward—or equivalently, to minimize regret—by learning an optimal policy. The statement in Section 4.2 that "we optimize causal effects rather than direct reward observations" refers to our estimation strategy, not a change in the underlying optimization objective. The reason we aim to maximize the causal effect rather than directly maximizing the cumulative reward is that the causal effect captures the fundamental differences between arms. This allows us to bypass non-stationary variations that arise not from the arms themselves, but from contextual information or confounding factors. To avoid potential confusion, we have revised the sentence to "We propose Dynamic Causal Bandit algorithms that optimize policies based on estimated causal effects, rather than potentially biased direct reward observations, in order to maximize cumulative rewards." This revision makes clear that reward maximization remains our objective, and causal effect estimation is employed as a means to address bias in the observed data.
>
> **Requested Change 3:**  We thank the reviewer for the helpful comment. We would like to clarify that we do not assume a linear reward-generating process. Assumptions 5 and 6 in Appendix D relate to the identification of causal effects, not to the reward mechanism itself. In particular, we write, "We model the treatment effect $\tau_{i,t}$ as a linear function of the context variables, denoted as $\tau_{i,t} = x_{i,t}^\prime \beta_i$, where $\beta_i$ is a vector of coefficients assumed to remain constant over time for each individual unit $i$." This modeling assumption applies only to the difference: $\tau_{i,t} := E\left[ R_{i,t}(1) - R_{i,t}(0) \mid H_{i,t-1} \right]$, i.e., the causal effect, not to the potential outcomes themselves. In other words, assuming that the causal effect is linear does not imply that the underlying reward function or the potential outcomes $R_{i,t}(1)$ and $R_{i,t}(0)$ are linear. We hope this clarification addresses the reviewer's concern.
>
> **Requested Change 4:**  We respectfully disagree with the reviewer. Our paper does not assume that treatment effects are stationary in all cases. In fact, we explicitly consider three different forms of non-stationarity, as described in the main text:
>
>
> 1. Time-invariant treatment effect (DCMAB):
> "In the case that the individual treatment effect remains \textbf{consistent} for each unit over time, we propose Dynamic Causal Multi-Armed Bandits (DCMAB), as outlined in Algorithm 1."
>
> 2. Context-dependent, time-invariant model (DCCB): "When contextual information $x_{i,t} \in R^{d}$ is available for each unit at each time point, we can extend our approach to Dynamic Causal Contextual Bandits (DCCB). We model the treatment effect $ \tau_{i,t}$ as a linear function of the context variables, denoted as $ \tau_{i,t} = x_{i,t}^{\prime} \beta_{i}$, where $\beta_{i}$ is a vector of coefficients assumed to remain constant over time for each individual unit $i$. "
>
>  3. Context-dependent, time-varying model (D-DCCB): "We model the treatment effect $ \tau_{i,t}$ as a linear function of the context variables, represented as $ \tau_{i,t} = x_{i,t}^{\prime} \beta_{i,t}$. Different from the model used in DCCB, here  $\beta_{i,t}$ is subject to change over time independently for each individual unit $i$. "
>
> These modeling options are designed to progressively relax assumptions about stationarity and allow more flexibility in capturing real-world dynamics. To prevent confusion, we have clarified these assumptions explicitly in the theoretical statements, as suggested. We also emphasize that even under the "constant-effect" setting, the observed rewards ($R_{i,t}(1)$ and $R_{i,t}(0)$ ) can remain non-stationary and nonlinear in the context, with unobserved heterogeneity and time-varying confounding still permitted (as discussed in our response to Request 3). We hope this clarification helps address any potential misunderstanding.

---

> > ### Author Response · Authors · 2025-06-22
> > **Point-by-point Response to Reviewer 8S5e Part II**
> >
> > **In response to the reviewer's broader concerns:** We respectfully disagree with the reviewer's assessment regarding the clarity and strength of our modeling assumptions. As outlined in Response 3 and Response 4, we have made careful efforts to clearly state our assumptions in the main text, with each modeling scenario presented in its own subsection or section. In particular, our assumptions are limited to the structure of the treatment effect, while allowing the reward functions to remain flexible and potentially non-stationary with latent variables. This design choice reflects a deliberate balance between modeling flexibility and identifiability, which is essential for theoretical analysis.  Notably, our work analyzes dynamic regret, where the benchmark is the optimal treatment at each time point, rather than a static optimal policy. To the best of our knowledge, no prior work under comparable settings achieves sublinear regret bounds - our work is the first to do so, which constitutes a key theoretical contribution.  Given these clarifications and the nature of our contributions, we believe that the abstract and introduction already accurately reflect the scope and key results of our work.
> >
> >
> >
> > Best,
> >
> > Authors of Paper4805

---

> > > ### Comment · Reviewer_8S5e · 2025-06-23
> > > **response**
> > >
> > > I thank the authors for the response, and sorry that the wording in my last response was not precise. I meant that the way the authors handle nonstationarity is by assuming that the treatment effect follows a factor model, which is very restrictive. I think it is very important for the authors to acknowledge this limitation in the introduction of their work, so it will not mislead the readers. I think the authors did not address my concerns over the writing of section 4.2: what are the assumptions that you need for (3) to be an unbiased estimate of (2)? The assumption numbers in thms 1 and 2 do not match those in the appendix. they need to be fixed. Additionally, the authors should discuss the role of these assumptions explicitly in the main body.

---

> > > > ### Author Response · Authors · 2025-06-26
> > > > **Point-by-point Response to Reviewer 8S5e II**
> > > >
> > > > We thank the reviewer for the clarification and constructive feedback. Please find our point-by-point responses below.
> > > >
> > > > **Regarding the assumptions about the treatment effect**
> > > >
> > > > We would like to clarify that our method does not assume the treatment effect follows a factor model. As outlined in our previous response to Requested Change 4, we consider three distinct models for treatment effects, which we restate here for completeness:
> > > >
> > > >
> > > > 1. Time-invariant treatment effect (DCMAB):
> > > > "In the case that the individual treatment effect remains consistent for each unit over time, we propose Dynamic Causal Multi-Armed Bandits (DCMAB), as outlined in Algorithm 1."
> > > >
> > > > 2. Context-dependent, time-invariant model (DCCB): 'When contextual information $x_{i,t} \in R^{d}$ is available for each unit at each time point, we can extend our approach to Dynamic Causal Contextual Bandits (DCCB). We model the treatment effect $ \tau_{i,t}$ as a linear function of the context variables, denoted as $ \tau_{i,t} = x_{i,t}^{\prime} \beta_{i}$, where $\beta_{i}$ is a vector of coefficients assumed to remain constant over time for each individual unit $i$. '
> > > >
> > > >  3. Context-dependent, time-varying model (D-DCCB): "We model the treatment effect $ \tau_{i,t}$ as a linear function of the context variables, represented as $ \tau_{i,t} = x_{i,t}^{\prime} \beta_{i,t}$. Different from the model used in DCCB, here  $\beta_{i,t}$ is subject to change over time independently for each individual unit $i$. "
> > > >
> > > >
> > > > We would like to clarify that our original introduction already states that we assume “either time-invariant treatment effects or linear treatment effect models,” and we have highlighted this sentence in blue for your convenience. As requested, we have also added the sentence “The proposed methods assume either time-invariant or linear treatment effect models, while accommodating flexible data-generating processes for potential outcomes” to the abstract. We emphasize that our model does not rely on a factor structure. Moreover, assuming a linear form for treatment effects is a widely accepted and standard practice in the contextual bandit and online causal inference literature. Importantly, our framework imposes minimal assumptions on the potential outcomes $R(0)$ and $R(1)$, allowing for substantial flexibility.
> > > >
> > > >
> > > > **Regarding the assumptions required for (3) to be an unbiased estimate of (2)**
> > > >
> > > >
> > > > We appreciate the reviewer’s thoughtful question. We would like to clarify that our approach does not rely on Equation (3) being an unbiased estimator of Equation (2) at all times. Instead, we provide an upper bound on the estimation error in Equation (9) of the appendix, which is adapted from the matrix completion literature. This result guarantees that the estimation error decreases as a function of the sample size $N$ and the round $t$.  Leveraging this bound allows us to establish the regret upper bound in our analysis.
> > > >
> > > > **Regarding the numbering and discussion of assumptions**
> > > >
> > > > Thank you for pointing out the numbering issue—we have corrected this in the revised version. To improve clarity, we have also moved a brief explanation of these assumptions from the appendix to the main text. For the reviewer’s convenience, we reproduce the explanation here: 'Assumptions A4 and A5 are standard boundedness conditions to prevent extreme reward values. Assumption A6 is a commonly used margin condition in the causal inference literature.'
> > > >
> > > >
> > > > Thank you again for your careful review and thoughtful feedback!
> > > >
> > > >
> > > > Best regards,
> > > >
> > > > Authors of Paper4805

---

### Decision · Action_Editor_vT7Y · 2025-08-01

**Recommendation:** Reject

**Audience:**

No

**Audience Explanation:**

In terms of methods, the two proposed algorithms are unsurprising, being based on straightforward plug-in model-based estimates with added epsilon-greedy steps. So that we have to consider only the regret bounds as a contribution that it specific to this submission. However, as detailed above, in their present form, the regret bounds obtained in the paper are unconvincing and rely on excessively restrictive ad hoc mathematical assumptions.

**Claims And Evidence:**

No

**Claims Explanation:**

The paper provides regret bounds for two epsilon-greedy strategies (for uncontextual and linear bandit) in a scenario which combines multiples agents and some form of non-stationary/inhomogeneity between agents. These bounds are sublinear (in time) thanks to instance-dependent tuning of the decrease of the epsilon greedy parameters. There is no corresponding lower bound for the analyzed scenarios.

The main concern raised by all reviewers is that the regret bound depend on very strong unnatural mathematical assumptions. The more questioned one being that N (the number of agents) is larger than T (the horizon). Reviewers also considered that the paper tends to overclaim its actual contributions and lacks clarity. This is mostly due to the fact that the results are stated in a very informal way in the main text, omitting most assumptions that can only be found in the supplementary material. Going back to the N > T one for instance, the exact assumption is A.10 in the supplementary material (page 5) which in fact has a set of four conditions linking N, T and other parameters which can only be satisfied under very stringent conditions on the values of N and T. For instance, Figure 4 (always in appendix) shows that for 200 time steps, it is required that N be at least twenty thousands for the conditions to hold. Always in appendix only, one finds two conditions A.8 and A.9 (together with A.7 implying a bounded conditions number for all design matrices at all times and for all agents) which is well known to be very strong in the context of linear bandits.

Despite relatively extensive feedback by the authors of this paper, the relevance and necessity of some of these assumptions could not be convincingly justified by the authors. Most reviewers also considered that despite the changes proposed by the authors the paper still remains very imprecise and does not adequately compare itself to the related literature on multi-armed bandits. In the end, all three reviewers recommend rejection of the paper.